# Microstructure and Mechanical Properties of In Situ Synthesized Metastable β Titanium Alloy Composite from Low-Cost Elemental Powders

**DOI:** 10.3390/ma16237438

**Published:** 2023-11-29

**Authors:** Krystian Zyguła, Tino Mrotzek, Oleksandr Lypchanskyi, Dariusz Zientara, Maik Gude, Ulrich Prahl, Marek Wojtaszek

**Affiliations:** 1Faculty of Metals Engineering and Industrial Computer Science, AGH University of Science and Technology, Mickiewicza Av. 30, 30-059 Kraków, Poland; oleksandr.lypchanskyi@imf.tu-freiberg.de (O.L.); mwojtasz@metal.agh.edu.pl (M.W.); 2Institute of Lightweight Engineering and Polymer Technology (ILK), Technische Universität Dresden, 01307 Dresden, Germany; tino.mrotzek@tu-dresden.de (T.M.); maik.gude@tu-dresden.de (M.G.); 3Institut für Metallformung, TU Bergakademie Freiberg, 4 Bernhard-von-Cotta-Straße, 09599 Freiberg, Germany; ulrich.prahl@imf.tu-freiberg.de; 4Faculty of Materials Science and Ceramics, AGH University of Science and Technology, Mickiewicza Av. 30, 30-059 Kraków, Poland; zientara@agh.edu.pl

**Keywords:** titanium composites, in situ reaction, powder metallurgy, hot compaction process, microstructure, strength properties

## Abstract

The titanium matrix composite was produced through a hot compaction process at 1250 °C using the mixture of elemental powders with chemical composition of Ti-5Al-5Mo-5V-3Cr and 2 wt.% addition of boron carbide. The phase analysis via X-ray diffraction method was performed to confirm the occurrence of an in situ reaction between boron carbide and titanium. Then, the wide-ranging microstructural analysis was performed using optical microscopy as well as scanning electron microscopy along with energy-dispersive X-ray spectroscopy and electron backscatter diffraction. Based on this investigation, it was possible to describe the diffusion behavior during hot compaction and possible precipitation capabilities of TiC and TiB phases. Tensile and compression tests were conducted to determine the strength properties. The investigated composite has an ultimate tensile strength of about 910 ± 13 MPa with elongation of 10.9 ± 1.9% and compressive strength of 1744 ± 20 MPa with deformation of 10.5 ± 0.2%. Observation of the fracture surface allowed us to determine the dominant failure mechanism, which was crack propagation from the reaction layer surrounding remaining boron carbide particle, through the titanium alloy matrix. The study summarizes the process of producing an in situ titanium matrix composite from elemental powders and B_4_C additives and emphasizes the importance of element diffusion and reaction layer formation, which contributes to the strength properties of the material.

## 1. Introduction

Titanium and its alloys exhibit significant potential for various applications, primarily due to their low density and unique combination of material properties [1]. β titanium alloys, in particular, have garnered considerable interest from the aerospace industry owing to their high strength, excellent fatigue properties, and outstanding corrosion resistance [2,3]. In recent years, titanium matrix composites (TMCs) have gained significant popularity. These composites, based on a titanium matrix, offer not only the aforementioned characteristics but also demonstrate good performance at elevated temperatures, high stiffness, and exceptional wear resistance [4,5,6,7]. The primary production technology for TMCs is powder metallurgy. This method enables the straightforward incorporation of reinforcing phases into the titanium alloy through an in situ reaction during powder sintering. It is nearly waste-free and can offer economic benefits by using inexpensive elemental powders [8,9,10] or even titanium chips, providing a method for recycling of metal scrap [11]. This approach allows for the production of near-net shape components, and the appropriate selection of process parameters ensures high density and strength properties.

The production of composites based on a titanium matrix using in situ powder metallurgy methods is currently a popular area of research [12,13]. The formation of in situ reinforcing phases is made possible by the high reactivity of titanium with carbon and boron [14]. Based on the exothermic reaction between Ti powder particles and the additional introduction of B_4_C particles into the powder mixture, two potential reinforcing phases can be obtained: TiB_x_ and TiC. As a result, the literature predominantly features research results related to the development of production parameters for these materials, as well as the characterization of their microstructure and basic strength properties [15]. These studies often focus on the reaction between pure Ti powder and commercially available B_4_C particles [16]. Published research results currently emphasize the determination of favorable sintering conditions that enable high composite density to be achieved and create a significant amount of additional reinforcing phase. Less frequently, research is conducted on the use of β titanium alloys as the matrix. In this field of research, attempts have primarily been made to use master alloy powders [17,18]. On the other hand, there is a lack of sufficient knowledge and research results concerning the behavior of multicomponent elemental powder mixtures. This approach holds immense significance for both research and industry. On one hand, there is a need to assess the potential of new materials through research activities that simultaneously investigate the economic process routes in an application-oriented manner based on the available parameters. The anticipated superior material properties compared to standard unreinforced alloys also serve as motivation to explore the relevant process parameters and material properties, with the aim of introducing new material systems for discussion across the entire mobility sector (aviation, automotive). Through appropriate post-processing strategies and functionalization possibilities, a wide range of applications can be created [19,20].

The in situ reaction that occurs during the sintering of powder particles leads to the precipitation of additional strengthening phases in the matrix. Achieving reinforcement in the form of TiB and TiC precipitate networks is possible through various powder metallurgy-based approaches. Wei et al. [21] used graphite powder and TiB_2_ powder as reinforcing additives. Through intensive ball milling and subsequent hot pressing, the powder particles of the reinforcing phases coated the Ti6Al4V alloy powder particles. This procedure, during sintering under high-pressure and high-temperature conditions, allowed graphite and TiB_2_ to react with the titanium matrix, forming a network of reinforcements along the boundaries of the primary Ti6Al4V powder particles. A similar manufacturing strategy for a TiAl alloy-based composite was employed by Pu et al. [22], who also used milled powders with the addition of B_4_C. The subsequent sintering process was conducted using the spark plasma sintering (SPS) method. Despite these differences, both cases achieved a similar effect in the form of a network of reinforcements at the boundaries of the primary powder particles of the alloy. Reinforcement in the form of randomly distributed precipitates evenly spread in the titanium matrix was attained through the mixing of Ti powder and B_4_C particles, as demonstrated in [23]. In this case, the authors attempted to use elemental Ti and Mo powders sintered using the SPS method at various temperatures. Despite achieving an in situ reinforcement effect and high-density, non-uniform areas rich in Mo were observed even when sintered at 1450 °C. This indicates that there is an area of research concerning multicomponent in situ TMCs that currently requires further investigation.

Considering the economic aspects, the preferred approach to producing TMCs involves manufacturing a semi-finished product using elemental powders and subsequently using a hot consolidation process, where a β titanium alloy serves as the matrix. This group of alloys is characterized by an excellent strength–weight ratio, and the microstructure can be modified through heat treatment and deformation [24,25,26]. Consequently, it allows for relatively easy control of the balance between high strength and ductility. Furthermore, powder metallurgy opens the opportunity for the easier incorporation of reinforcing phases in the form of ceramic B_4_C particles into the chemical composition. The appropriate choice of manufacturing parameters influences the efficiency of in situ reactions, during which additional TiB and TiC phases precipitate. This provides the opportunity to control thermal properties and enhance strength.

The consolidation conditions have a crucial impact on the microstructure of the resulting product, which directly affects its strength properties [27,28]. In general, lamellar structures in titanium alloys are characterized by high strength and toughness but low ductility [29]. Presently, most research on the synthesis of in situ TMCs is conducted using commercially available pure titanium powder (CP-Ti) and B_4_C particles. In [30,31], the effect of the addition of B_4_C on the microstructure and properties of a CP-Ti matrix composite obtained through SPS was analyzed. The material was heated and sintered in the β-phase region. The resulting microstructure of this composite consisted of massive α-phase lamellae, suggesting a relatively slow cooling rate. It is worth noting that pure titanium exhibits low strength. For instance Attar et al. [32] investigated the mechanical behavior of pure Ti and Ti-TiB composite materials manufactured using selective laser melting. They demonstrated a compressive strength of the TiB-added composite at approximately 1420 MPa. Additionally, in [33], it was shown that the tensile strength and elongation decrease with an increasing amount of the reinforcing phase. Therefore, using CP-Ti as a starting material is a suitable approach for studying the reaction kinetics between titanium and reinforcing phase particles. Controlled microstructure morphology can only be achieved through the addition of alloys to the matrix material and increasing its strength. In the research presented in [34], Ti-5553 alloy powder and B_4_C particles were employed. The sintering process was conducted at a temperature above the α + β → β phase transformation, with an uncontrolled cooling process. As a result, the matrix microstructure contained β-phase grains with equiaxed morphology, suggesting relatively rapid cooling. The use of the alloy as the matrix material led to an improvement in strength properties, with the compressive strength increasing to 1530 MPa for the mixed powder composite and up to 2450 MPa for the milled powder composite.

In the present study, the primary research objective was to develop an approach for synthesizing an in situ titanium matrix composite from the multicomponent elemental powder mixture. Special attention was given to the production process involving hot compaction, using cost-effective elemental powders with a chemical composition corresponding to the Ti-5Al-5Mo-5V-3Cr alloy, with the addition of 2 wt.% of B_4_C. This study places a strong emphasis on providing a comprehensive analysis of the underlying phenomena during the hot compaction of these powders and subsequently the fracture behavior exhibited by the resulting composite. The research methodology allowed for detailed examination of the phase composition of the in situ TMC, coupled with analysis of the microstructure obtained through the hot compaction process. Furthermore, the mechanical properties of the TMC, specifically focusing on both tensile and compressive strength. To gain insights into the failure mechanism, investigation extends to the examination of fracture surfaces.

## 2. Materials and Methods

The procedure for preparing the powder mixture has been extensively detailed in [35]. In general, the initial material consisted of elemental powders of titanium, aluminum, molybdenum, vanadium, chromium, and boron carbide. The mixture was prepared by weighing the elemental powders in proportions that corresponded to the chemical composition of the Ti-5Al-5Mo-5V-3Cr (Ti-5553) alloy, supplemented with 2 wt.% of B_4_C (approximately 3.58 vol.%). Subsequently, the powders were blended within a ceramic mixing chamber, using 8 mm diameter tungsten carbide balls. The mass ratio of the balls to the mixture was 1:1, and the mixing process persisted for a duration of 90 min at a mixer speed of 55 rpm.

The hot compaction process was executed using a 25 kN press (Thermal Technology Press Inc., Minden, NV, USA) at a temperature of 1250 °C for a duration of 1 h within an argon protective atmosphere. The compaction pressure applied was 25 MPa, resulting in the compression of the powder mixture into a cylindrical powder compact, measuring 78 mm in diameter and 42 mm in height. Subsequently, the compact was gradually cooled within a furnace to reach room temperature. The measured relative density was found to be 4.65 ± 0.01 g/cm^3^. In Figure 1, the schematic representation of the processing route for in situ composite production from elemental powders is presented.

X-ray diffraction (XRD) analysis was conducted utilizing a Panalytical Empyrean DY 1061 (Malvern Panalytical, Malvern, UK), X-ray diffractometer, equipped with a Cu lamp emitting Kα radiation at a wavelength of 1.54 Å. The analysis covered an angular range of 2θ from 20° to 90°, with a scanning step size of 0.03° and a scanning frequency of 7 s, operating at 40 kV and 40 mA. For microstructural examination, specimens were prepared following a standard grinding and polishing procedure, followed by etching with Kroll reagent, which comprised 2% HF, 6% HNO_3_, and 92% H_2_O. Microstructural observations were conducted employing a Leica DM400M light microscope (Leica Microsystems GmbH, Wetzlar, Germany), as well as Hitachi TM-3000 (Hitachi, Ltd., Tokyo, Japan) and FEI Inspect S50 (FEI Company, Hillsboro, OR, USA) scanning electron microscopes, both equipped with an energy-dispersive spectrometry system (EDS). Additionally, electron backscatter diffraction (EBSD) analysis was performed using a Zeiss GeminiSEM 450 scanning electron microscope (Carl Zeiss AG, Oberkochen, Germany), operating at a working voltage of 20 kV and utilizing a step size of 0.2 μm.

Tensile tests were conducted using a tensile testing module supplied by Kammarth Weiss GmbH (Schwerte, Germany) using dedicated flat tensile specimens and test speed of 0.005 mm/s. Compression tests were executed in accordance with the ISO 13314:2011 standard [36] using a Zwick-Roell Z250 testing machine (Zwick-Roell GmbH & Co. KG, Ulm, Germany), where cylindrical specimens measuring 6 mm in diameter and 9 mm in height were employed. The compression tests were carried out at constant strain rate of 0.05 1/s, accompanied by a preload of 1500 N. The compression curves were corrected using the compliance curve approach. The tensile and compression tests were performed 3 times each.

## 3. Results

### 3.1. Phase Identification and Microstructure Observations

The XRD patterns of the TMC and the Ti-5553 alloy used as a reference are presented in Figure 2. In the case of the Ti-5553 alloy, which serves as the matrix material for the TMC, the hexagonal Ti_α_ and body-centered cubic Ti_β_ phases were identified. The presence of these two phases is characteristic of a metastable β titanium alloy that experienced slow cooling from above the phase transition temperature to room temperature [37,38]. An examination of the TMC specimen, in addition to the phases observed in the matrix material, revealed the presence of multiple phases that precipitated through in situ reactions during the hot compaction process. These phases were identified as TiB, TiB_2_, TiC, and B_4_C. The existence of these phases suggests the occurrence of in situ reactions, as governed by Equation (1) during the hot compaction process. Certain XRD peaks indicate the presence of the TiB_2_ phase, suggesting the possibility of reaction (2). The Gibbs free energy (ΔG) for these reactions is below zero, indicating their feasibility within the temperature range of 800–1800 °C [39,40,41], which makes them feasible under the hot compaction conditions applied during the consolidation of the investigated TMC. Furthermore, Yi et al. [39] demonstrated that the ΔG values for both reactions are similar, meaning that the occurrence of both reactions is equally feasible. Peaks from the B_4_C phase were also observed, indicating that not all the boron carbide particles introduced into the mixture reacted completely.
5Ti + B_4_C = 4TiB + TiC (1)
3Ti + B_4_C = 2TiB_2_ + TiC (2)

Figure 3 illustrates the optical microstructures of a polished and etched specimen. The microstructure of the matrix comprises colonies of thin α-phase laths positioned within the β-phase matrix and massive α-phase lamellae dispersed along the primary β grain boundaries. This particular microstructure, observed in metastable titanium alloys such as the Ti-5553 alloy, forms during slow continuous cooling from a temperature exceeding the β-transus point. After the hot compacting process, the material was cooled together with the furnace from a temperature of 1250 °C. During cooling, the primary α phase grains initially precipitate at the boundaries of pre-existing β phase grains and subsequently undergo considerable growth. As a result, they adopt the configuration of long and massive lamellae, commonly referred to as grain boundary α (α_GB_). Within the β-phase grains, nucleation of primary α-phase precipitates occurs in the form of thin laths that arrange themselves in colonies. The presence of α_GB_ grains hinders the further growth of the primary α grains within the original β grains, giving rise to the formation of enclosed clusters of primary α grains. The darker regions denote colonies of secondary α-phase grains, which manifest as very thin needles.

Figure 3A presents a retained B_4_C particle surrounded by a reaction layer. The boron carbide particle is encircled by a void and lacks a continuous connection to the matrix; instead, it connects to the reaction layer via diffusion necks. This is the only significant porosity that has been observed. The localized low consolidation in the neighborhood of the B_4_C particles primarily arises from the exceptionally high melting point of these particles (2350 °C), far exceeding the temperature employed during hot compaction. Consequently, atom migration into the matrix occurred through a solid-state diffusion process, with the surrounding reaction layer effectively impeding reverse diffusion. The reaction layer forms a direct link between the introduced strengthening phase and the Ti-5553 alloy matrix, facilitating the diffusion of elements such as boron and carbon. The average reaction layer thickness was determined through microscopic observations, relying on six measurements. Two measurements of the reaction layer were taken on each side of the remaining B4C particle, with an additional measurement at both the top and bottom of the visible particle. This process was applied to five selected particles. The average value of the reaction layer was 45.3 ± 7.5 µm. At the boundary of the reaction layer, whiskers of the precipitated strengthening phase are discernible. Figure 3B displays a uniformly distributed colony of precipitated strengthening phases. Within this colony, both plate-like phases and elongated whiskers and/or blocks are evident, preferentially precipitating at the grain boundaries of the primary β-phase. Further identification of these phases was conducted through SEM-EDS analysis.

In order to distinguish and identify individual phases within the microstructure, SEM-EDS analysis was conducted. SEM images that feature phases identified through point EDS analysis are presented in Figure 4. Figure 4A shows a retained B_4_C particle surrounded by a reaction layer. Point analysis indicated that the B_4_C particle predominantly contained boron and carbon, while the reaction layer primarily consisted of titanium, boron, with a lesser amount of carbon, and addition of molybdenum and vanadium. The application of the hot compaction process facilitated the development of a continuous interface between the reaction layer and the matrix, enhancing the diffusion process and resulting in the formation of a relatively thick reaction layer. It is expected that, as the time and/or temperature of the hot compaction process increases, the thickness of the reaction layer will also increase at the expense of the remaining B_4_C particle, due to the prolonged time of elemental diffusion and greater mobility of atoms at elevated temperatures. At the edges of the reaction layer (Figure 4B), strengthening phase precipitates in the form of TiC particles and TiB whiskers or blocks were observed. TiC particles concentrated primarily along the primary β phase boundaries in the α_GB_ region or as colonies of massive blocks (Figure 4C). The TiB phase predominantly appeared as long transgranular whiskers (Figure 4D) or blocks, which were also carbon enriched.

The hot compaction process occurs without transitioning into a liquid phase, leading to the formation of new phases through solid-state reactions governed by diffusion. When elemental powders are employed, the diffusion process is further hindered by the presence of naturally occurring oxide layers on the surface of the powder particles. Throughout the process, the material experiences gradual heating from the outer surface to the interior. In the context of a combined consolidation and high-temperature sintering process, the oxide layers are initially disrupted as a result of the movement and deformation of powder particles. Subsequently, with the elevation of temperature, these oxide layers dissolve, thereby enabling the diffusion of atoms.

Figure 5 displays EDS elemental distribution maps, revealing distinct concentration patterns. In particular, the α_GB_ phase exhibits a predominant concentration of titanium and aluminum, as aluminum functions as a stabilizer for the α phase in titanium alloys. Conversely, the other alloying elements, such as molybdenum, vanadium, and chromium, which serve to stabilize the β phase, are uniformly distributed within it. Boron is concentrated solely within the B_4_C particles, the reaction layer, and the TiB phase; meanwhile, carbon naturally displays higher concentrations in the TiC particles but is also evenly distributed in the Ti-5553 alloy matrix. In Figure 5a, the retained B_4_C particle encircled by a reaction layer is presented. Besides the high concentrations of boron and carbon within the retained B_4_C particle, there is a minor presence of molybdenum and chromium along the particle’s edge. As previously indicated, the reaction layer exhibits elevated concentrations of titanium, boron, molybdenum, and vanadium but relatively lower levels of carbon. Especially, aluminum and chromium are absent within this layer. It is worth mentioning the numerous TiC particles precipitating at the boundary of the reaction layer. Figure 5c presents an EDS map of a cluster of massive TiC phase particles. Within these particles, a significant concentration of carbon and titanium is evident, while aluminum, molybdenum, and chromium are absent.

### 3.2. Mechanical Properties

The ultimate tensile strength (UTS) of the investigated composite is summarized in Figure 6a. The UTS is found to be 910 ± 13 MPa, with an elongation of 10.9 ± 1.9%. There is no distinct yield point observed during tensile testing. Yang et al. [42] examined the tensile strength of Ti-5553 alloy, also produced from elemental powders, and reported a UTS of approximately 800 MPa after hot compaction. Therefore, the composite developed in this study exhibits an enhanced tensile strength compared to the matrix alloy. Similar conclusions were drawn by Wang et al. [43], who demonstrated that the addition of both B_4_C and C, which react in situ with the Ti-1100 alloy matrix to create additional strengthening phases, leads to an increase in UTS from 6.1 to 18.9% in comparison to matrix alloy. The reason for the observed increase in tensile strength for the titanium matrix composite with B_4_C addition was demonstrated by Jia et al. [44], who investigated how UTS changes depending on the size of B_4_C particles added to pure Ti powder. There is a correlation between tensile strength, elongation, and the size of B_4_C particles. The addition of fine B_4_C results in an increase in strength from 622 MPa to 1117.76 MPa but a significant decrease in elongation from 28.28% to 6.9%. With an increase in the size of B_4_C particles, strength decreases significantly, while elongation slightly increases. It was indicated that the main factor determining the increase in UTS is the strengthening from in situ-formed reinforcements. It has been shown that, when using large B_4_C particles, the in situ reaction does not fully occur, resulting in fewer TiC and TiB phases and hence less reinforcement from in situ formed reinforcements. It can be assumed that, by controlling the addition of the reinforcing phase, it is possible to control the strength properties of in situ titanium matrix composites. Furthermore, relatively high elongation values were achieved during tensile tests. It is widely accepted that powder metallurgy-processed titanium alloys exhibit lower elongation during tensile testing than their counterparts produced using conventional methods, with the morphology of the α phase being a key factor in this regard [45]. The microstructure of the Ti-5553 alloy after slow cooling from the β phase range consists of lamellae of the α phase arranged in colonies within the β phase matrix, which is generally an unfavorable microstructural state in terms of elongation. Consequently, the most commonly employed heat treatment for these alloys is aging at an appropriate temperature, which yields a balance between UTS and relatively high elongation [46]. Additionally, Fattahi et al. [30] demonstrated that in situ titanium-based composites produced from powders are particularly susceptible to early crack initiation in areas with increased porosity and due to weak interfacial bonds between the matrix and the reinforcing phase additives. Nevertheless, microstructure observations of the TMC examined in this study revealed the presence of a solid and continuous interface between the reaction layer and the matrix alloy, thanks to the method of hot compaction employed for consolidating elemental powders. This interface might be the primary reason for the achieved high elongation.

As anticipated, a high compressive strength was achieved (1744 ± 20 MPa), with an average compressive yield strength of approximately 1396 ± 39 MPa (Figure 6b). Compared to the unreinforced material, a slightly lower value of compressive strength (1894 MPa) was shown [47], but this was about 200 MPa higher than that for the matrix alloy obtained by casting and heat treatment [48]. Grützner et al. [49] investigated the compressive strength of a Ti-5553 alloy with the addition of 4 vol.% of B_4_C, produced through the spark plasma sintering method. They reported a nearly identical compressive strength (around 1780 ± 30 MPa), slightly lower compressive yield strength (approximately 1190 ± 30 MPa), and higher deformation (17.7 ± 1.1). These variations may be attributed to the alloy powder used and the attainment of a more homogeneous microstructure in terms of chemical composition. Nevertheless, the similar results obtained indicate that this is another proof that by controlling the addition of the reinforcing phase, it is possible to control the strength properties of in situ titanium-based composites.

## 4. Discussion

### 4.1. In Situ Reaction and Diffusion Behavior

To gain a better understanding of the influence of in situ reactions occurring during hot compaction on the microstructure, EBSD maps of two characteristic regions were prepared. Figure 7 shows an area near an unreacted B_4_C particle. The inverse pole figure (IPF) maps were divided based on the indexed titanium phase. Figure 7a shows the grains of the β phase, while Figure 7b displays the grains of the α phase. While knowing that the α_GB_ phase is formed at the primary grain boundaries of the β phase, the IPF maps provide an estimate of the grain size of the high-temperature β phase. It can be seen that, due to the high temperature of hot compaction, primary β grains have grown to considerable sizes, exceeding 100 µm. Only a slight misorientation is apparent for some β phase grains. The alignment of α_GB_ phase grains on the primary β phase boundaries is clearly visible, as well as the primary α phase grains emerging from these boundaries into the interior of the β phase. It is noteworthy that the unreacted B_4_C particle, as well as the reaction layer, remain unindexed. By employing band contrast maps and simultaneous EDS analysis, it was possible to identify the phases that precipitated in situ. It was observed that TiB particles and whiskers also remained unindexed, while the TiC phase was indexed along with the cubic β phase, as both phases possess the same crystallographic structure. The second characteristic area exhibits colonies of TiB and TiC precipitates (Figure 8). The primary grains of the β phase are also coarse. In Figure 8b, distinctly visible long grains of the α_GB_ phase restrict the primary grains of the β phase and exhibit random orientation. On the other hand, colonies of primary α grains have the same orientation within one colony. Additionally, within one former β grain, more than one colony of primary α can be observed, and these colonies have different crystallographic orientations within one grain of the β phase. Similarly, through EDS analysis, it is evident that TiC particles and TiB whiskers are clearly identified. It was observed that the TiC phase exhibits random orientations and is located not only at the boundaries of primary β phase grains but also manifested intergranularly.

To examine and describe how in situ reactions occur during the hot compacting of a mixture of elemental powders with the addition of B_4_C, a linear EDS analysis was conducted. This analysis helps in better understanding the diffusion process of elements from the particle to the matrix. Figure 8 presents the results of the linear EDS analysis in two characteristic areas: line scan I, between the B_4_C particle and the reaction layer (Figure 9a), and line scan II, between the reaction layer and the matrix (Figure 9b). Figure 9c also illustrates an SEM image of the area where the measurement was conducted, with lines indicating the paths along which the EDS analysis was performed.

Naturally, the concentration of boron and carbon in the unreacted particle is higher compared to other elements. At the boundary between the particle and the reaction layer, there is a noticeable increase in chromium content. This observation was also confirmed by EDS mapping (Figure 5a), where a higher concentration of Cr, as well as C and Al, is visible at the edge of the retained B_4_C. The complexity of the chemical composition of the mixture of elemental powders means that, in addition to the expected reaction of titanium with boron carbide, reactions between other alloying elements and boron or carbon may also occur. Therefore, the reverse direction of diffusion is possible, i.e., from the matrix through the reaction layer into the edge of the unreacted particle, for elements such as Cr and Al, which form respective carbides or borides at the particle boundary. In Figure 4A and Figure 5a (BSE image), a phase with a slightly different color than the reaction layer can be seen at the location of Cr concentration at the edge of B_4_C. The concentration of other elements (Ti, Mo, and V) within the B_4_C particle is very low. In situ reactions are dependent on the diffusivity of different elements relative to other alloying elements and on the sintering process conditions, such as temperature and time. The issue of reactions between aluminum and boron is well understood [50], making it possible to provide a detailed explanation of the diffusion of aluminum into the boron particle. In general, the formation of new phases in the Al-B system is primarily dependent on three factors: chemical composition, temperature, and the duration of the process. The reaction between aluminum and B_4_C begins at around 700 °C, which occurs at a very early stage when the reaction layer is not yet fully formed. However, the complete reaction between aluminum and boron carbide requires prolonged exposure to elevated temperatures. In the case of the investigated alloy, with increasing temperature, the diffusion of other alloying elements becomes more significant, particularly titanium, which is in excess, thus inhibiting further reaction between Al and B. Chromium also exhibits a significant affinity for boron and carbon, and its reaction with B_4_C shows Gibbs energies below 0, making it possible to occur during the hot compaction process [51]. However, as demonstrated in [31,41,52], the reaction based on Equation (1) has a significantly lower Gibbs energy than the reaction of Cr or Al with B_4_C. Therefore, this reaction will be favored at high temperatures.

The reaction layer mainly consists of titanium and boron, with a noticeable decrease in carbon content. The percentage of boron relative to titanium suggests that it is the TiB phase. However, Nartu et al. [53] demonstrated that, during in situ reactions in the reaction layer, particles of the TiB_2_ phase also precipitate. The conducted research also appears to confirm this phenomenon. The presence of the TiB_2_ phase was confirmed in XRD studies, and linear EDS analysis indicates local increases in B content with corresponding decreases in Ti content. In the reaction layer, a relatively high content of V and Mo was observed, which is comparable to the concentrations of these elements in the matrix. The presence of other alloying elements in the reaction layer may hinder further diffusion of boron into the titanium alloy matrix, as can be observed in the comparative analysis of element concentrations between the reaction layer and the matrix, where the B content decreases. Furthermore, it has been previously demonstrated that aluminum can also hinder the diffusion of boron in titanium [54]. The concentration of other alloying elements, such as Al and Cr, increases in the matrix, and their fluctuations mainly depend on the presence of a specific titanium phase, as Al is a stabilizer of the α phase, and Cr of the β phase. The same applies to Mo and V. On the other hand, the carbon content in the reaction layer and matrix remains essentially unchanged. Therefore, it can be assumed that, due to the high affinity of carbon for titanium and other alloying elements, carbon diffuses much faster than boron, which is retained in the reaction layer, partly due to its low diffusivity in TiB_2_, a component of this layer [55]. Hence, microstructural observations showed a lower occurrence of TiB particles compared to numerous TiC precipitates.

Understanding the above phenomena allows for the proposing of a scheme for preparation of the in situ composite from elemental powders with the addition of B_4_C during the hot compaction process, as presented in Figure 10. In the first stage, the mixture of elemental powders based on titanium with the desired chemical composition is enriched with B_4_C particles. In the early stages of the hot compaction process, pressure mechanically connects the particles, causing their deformation, or in the case of hard B_4_C particles, fragmentation. This results in a green compact. Subsequently, as a result of temperature and time, B and C diffuse into the matrix, leading to reactions between B_4_C and titanium. A thick reaction layer forms on the surfaces of the retained B_4_C particles and the TiC and TiB phases precipitate, strengthening the material. During the hot compaction process, at the interface of B_4_C and the Ti alloy matrix, boron and carbon atoms initially decomposed, forming the beginning of the reaction layer. Subsequently, with increasing temperature and time, these atoms diffused into the Ti alloy matrix, creating TiB and TiC phases. The Gibbs free energy for the reaction between Ti and C is lower than that for the reaction between Ti and B. Therefore, as mentioned earlier, the formation of TiC was more preferred during the hot compaction process. Additionally, the carbon diffusion rate into the Ti alloy matrix is higher, resulting in a higher concentration of C at the edge of the reaction layer, as clearly seen in Figure 4B. This figure also illustrates the accumulation of TiC phases, forming a coherent part of the reaction layer. As the holding time at temperature increased, two main phenomena occurred. The first was the formation of TiB phase whiskers and the growth of existing TiC phases at the edge of the reaction layer. Simultaneously, the continued diffusion of carbon and boron atoms into the Ti alloy matrix led to the formation of additional TiC phases and TiB whiskers randomly spread on the titanium alloy matrix. The growth of these in situ precipitated phases at the edge of the reaction layer limited the diffusion of B and C atoms, resulting in incomplete reactions and the presence of retained B_4_C particles. This is how the in situ composite on a titanium alloy matrix, known as TMC, is created.

### 4.2. Fracture Behavior of TMC

To conduct a more in-depth analysis of the mechanical test results and establish the relationship between in situ reinforcement and strength properties, fracture analysis was performed. SEM micrographs of the fracture surface after the tensile test are shown in Figure 11. In Figure 11a, a fractured B_4_C particle with its interior exposed is visible. It is surrounded by a reaction layer with clear cracks. Additionally, TiB whiskers are clearly visible, and while some minor cracks are present, they are generally well preserved (Figure 11b,d). In Figure 11b, besides the marked cracks on the reaction layer, it can be seen the spot where a B_4_C particle was pulled off. In Figure 11c, the retained B_4_C particle in the center is visible, with half of its surrounding reaction layer pulled off. Basically, for all observed reaction layers surrounding the B_4_C particles, whether they are intact, fractured, or pulled off, propagating cracks can be seen. The observation of the fracture surface of the titanium alloy matrix revealed the predominant presence of ductile dimples with some cleavage facets, which essentially confirms significant elongation during the tensile test. A similar mechanism was observed for in situ composites on a pure titanium matrix with the addition of B_4_C, produced by the SPS method; although, in that case, brittle cracking predominated [41].

The analysis of the cracking mechanism in TMC also involved the examination of the microstructure near the surface of the longitudinal section of the specimen after the tensile test, as shown in Figure 12. The fractured B_4_C particles are presented in Figure 12a,c, and they are surrounded by a reaction layer with numerous cracks visible. In Figure 12b, a pull off portion of the reaction layer at the fracture surface is visible, exposing the surface of the B_4_C particle. As can be seen in Figure 12b,c, the cracks in the reaction layer propagate to the B_4_C particle and further into the titanium alloy matrix. In Figure 12d, a distinct crack is visible underneath the precipitation of the TiC phase, along with some minor cracking through it. Due to the low number and small size of observed pores in the microstructure, they were not considered a significant factor in initiating cracks in the tensile test. Observations of the microstructure near the fracture surfaces confirmed the earlier fractographic analysis.

The comprehensive fractographic analysis allows for the description of the dominant cracking mechanism in TMC produced through the hot consolidation process. This mechanism involves the cracking of the reaction layer, followed by the propagation of the crack towards the matrix and the B_4_C particles. The TiB phase present in the reaction layer has a lower elastic modulus compared to the matrix alloy, so it bears more of the tensile load, causing it to crack initially. The more ductile matrix initially suppresses the crack propagation, allowing for the continued transfer of the load through TiB until the specimen eventually fractures [56].

## 5. Conclusions

As part of the research conducted the in situ TMC was produced using hot compaction process. Microstructure, phase composition, and mechanical properties were investigated. The diffusion behavior and fracture mechanism were analyzed and discussed in detail. The following conclusions can be drawn:The study successfully demonstrated the in situ formation of a TMC material through a hot consolidation process for the composition of elemental powders with B_4_C additives. The applied process parameters made the occurrence of a reaction between Ti and B_4_C possible; as a result, additional strengthening phases precipitated.The microstructure of the TMC was found to be complex, consisting of various phases, including TiB, TiB_2_, TiC, and retained B_4_C. These phases formed as a result of solid-state reactions during hot compaction, contributing to the composite’s properties.The TMC exhibited beneficial mechanical properties, with a tensile strength of approximately 910 ± 13 MPa with elongation 10.9 ± 1.9% and compressive strength of 1744 ± 20 MPa with deformation of 10.5 ± 0.2%. The presence of multiple strengthening phases contributed to the increased tensile strength of the composite compared to the pure matrix alloy.Fractographic analysis revealed that the dominant cracking mechanism in the TMC was the cracking of the reaction layer, followed by crack propagation into the matrix and B_4_C particles. The TiB phase in the reaction layer played a crucial role in bearing the tensile load.The study outlined a comprehensive process for producing in situ composites from elemental powders with B_4_C additives, emphasizing the importance of the diffusion of elements and the formation of a reaction layer, which contributes to composite strength properties. This study not only enhances our understanding of the current state of TMC manufacturing but also reveals directions for further research. Understanding the details of element diffusion kinetics will enable the exploration of manufacturing parameter combinations and its influence on material properties, as well as the application of different elemental powder mixtures. Moreover, these directions in future research have the potential to develop the research field further and contribute to the application of TMCs in various industrial sectors.

## Figures and Tables

**Figure 1 materials-16-07438-f001:**
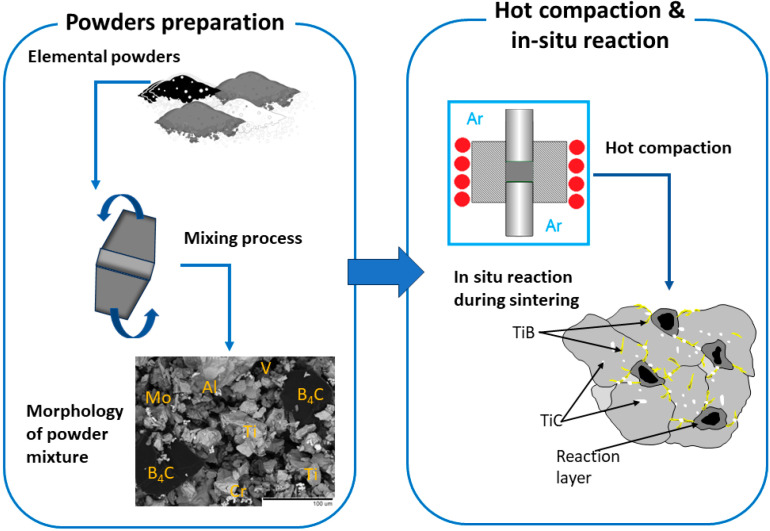
Schematic representation of the processing route for in situ composite production from elemental powders.

**Figure 2 materials-16-07438-f002:**
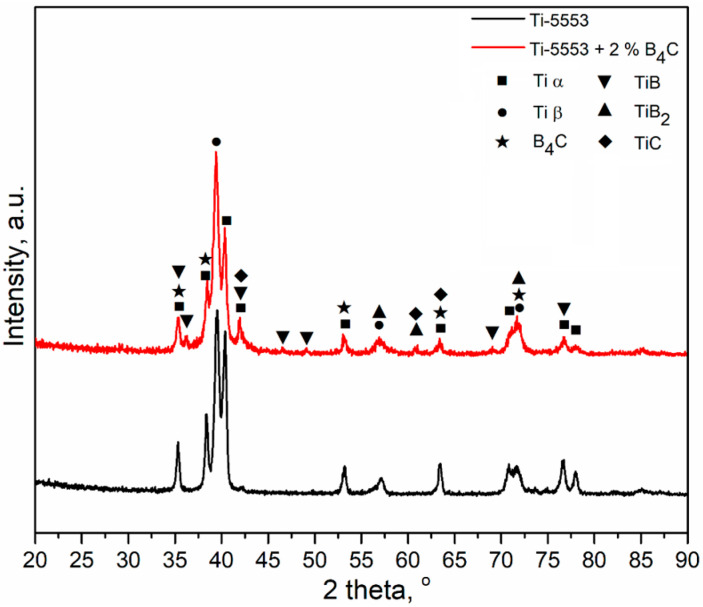
XRD patterns of the as-hot-pressed Ti-5553 alloy and TMC.

**Figure 3 materials-16-07438-f003:**
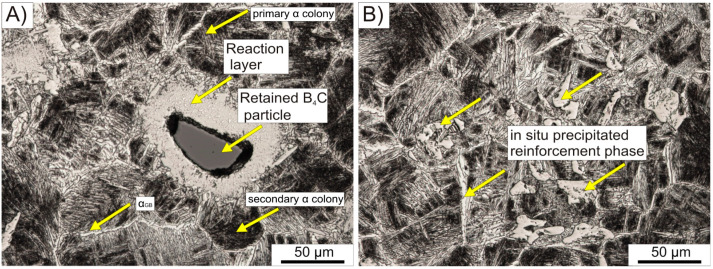
Optical microstructures of hot compacted Ti-5553 + 2%B_4_C composite: (**A**) view of retained B_4_C particle; (**B**) in situ precipitated TiC and TiB colony.

**Figure 4 materials-16-07438-f004:**
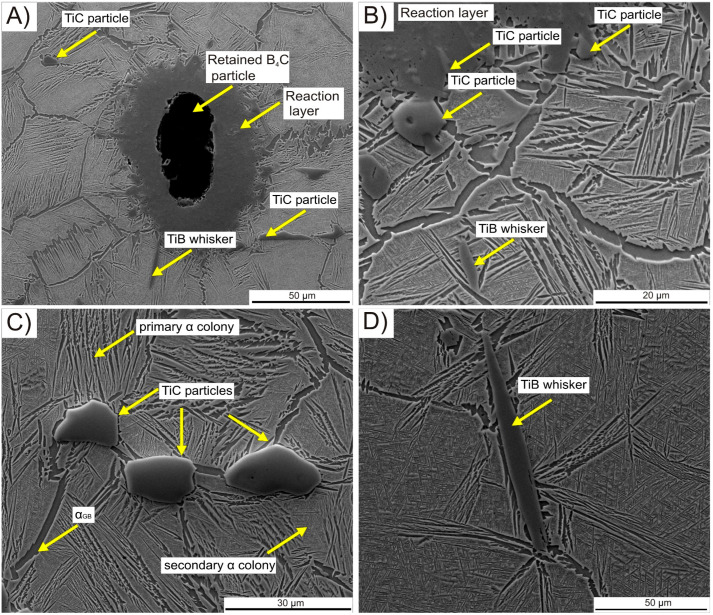
SEM images of the microstructure of TMC: (**A**) retained B_4_C particle; (**B**) edge of the reaction layer; (**C**) TiC particles on the Ti-5553 alloy matrix; (**D**) TiB whisker.

**Figure 5 materials-16-07438-f005:**
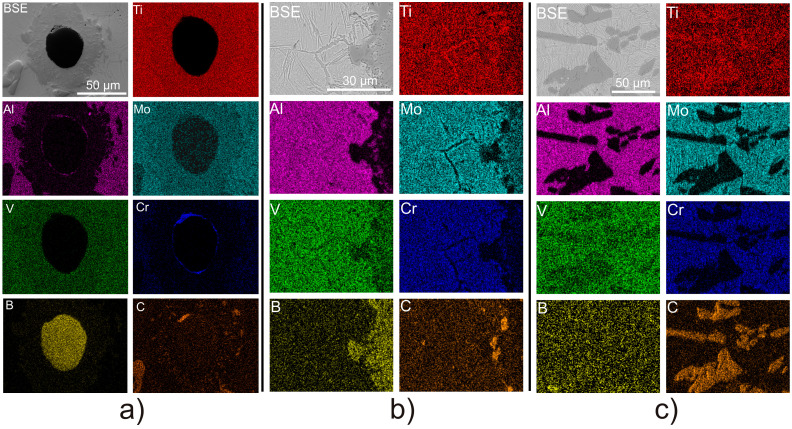
EDS mapping of (**a**) retained B_4_C particle; (**b**) interface between the reaction layer and Ti-5553 matrix; (**c**) in situ precipitated TiC particles colony.

**Figure 6 materials-16-07438-f006:**
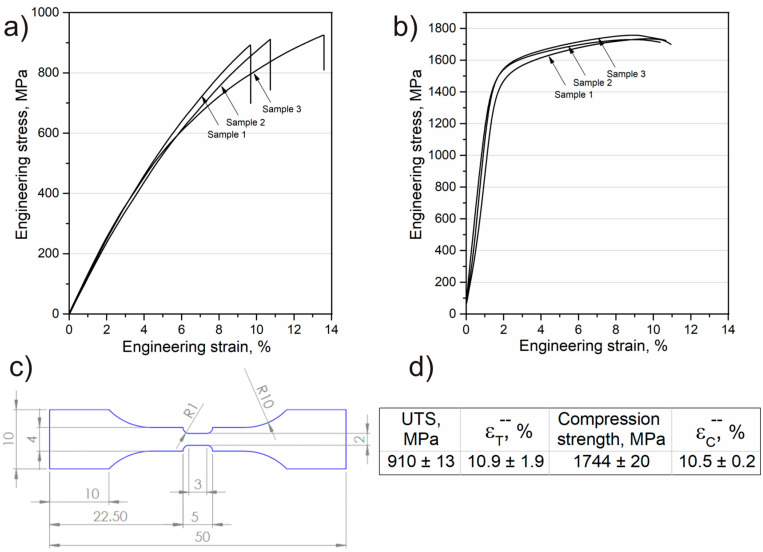
(**a**) Engineering stress–engineering strain curves for tensile and (**b**) compression tests; (**c**) schematic drawing of the tensile specimen (dimension in mm); (**d**) average values of strength properties.

**Figure 7 materials-16-07438-f007:**
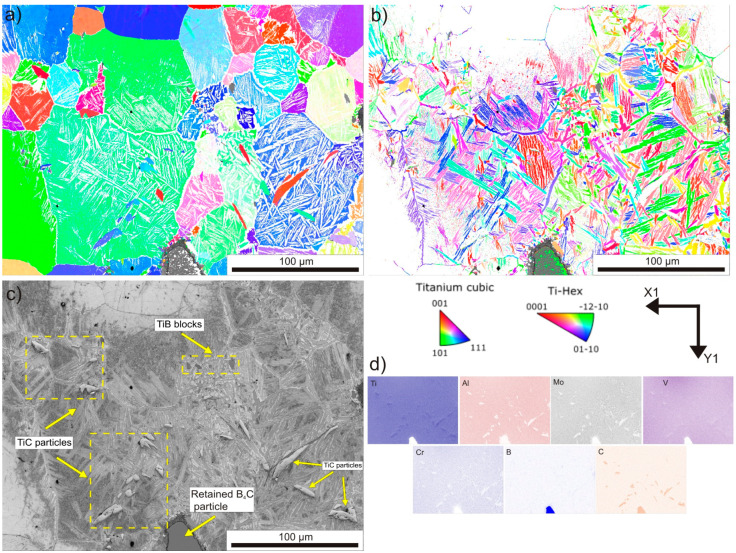
The EBSD maps of the area near the retained B_4_C particle: (**a**) IPF map of β phase; (**b**) IPF map of α phase; (**c**) band contrast map; (**d**) EDS chemical element maps.

**Figure 8 materials-16-07438-f008:**
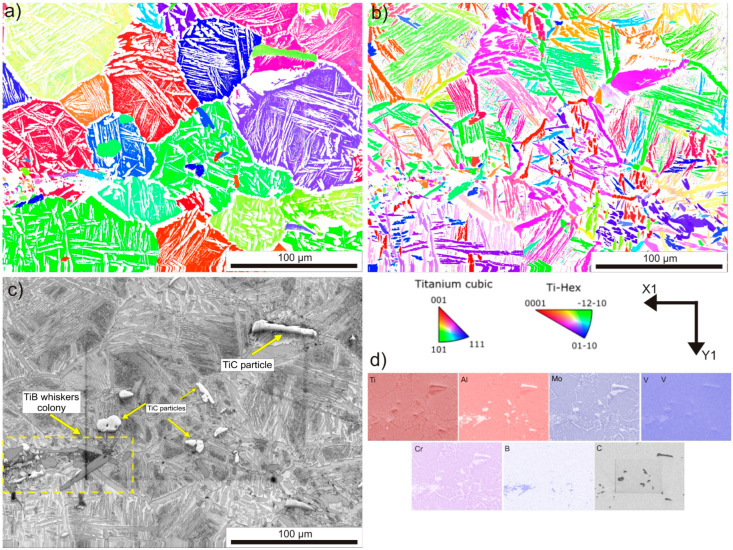
The EBSD maps of the colony area of the TiC and TiB precipitations: (**a**) IPF map of β phase; (**b**) IPF map of α phase; (**c**) band contrast map; (**d**) EDS chemical element maps.

**Figure 9 materials-16-07438-f009:**
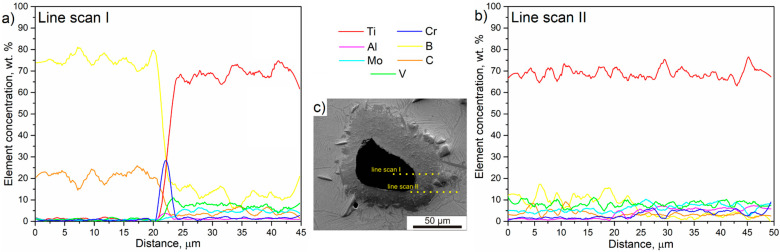
(**a**,**b**) EDS line scan measurements results of TMC. (**c**) SEM image of retained B_4_C particle with marked corresponding EDS line scans.

**Figure 10 materials-16-07438-f010:**
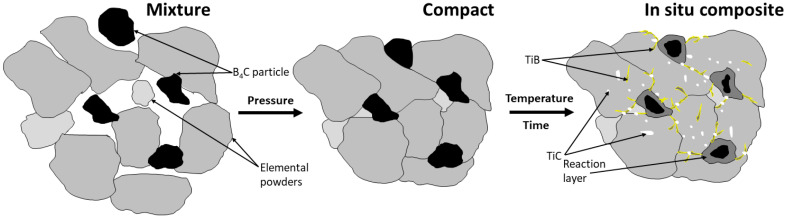
Schematic of preparation of in situ titanium-based composite from elemental powders with B_4_C addition during hot compaction process.

**Figure 11 materials-16-07438-f011:**
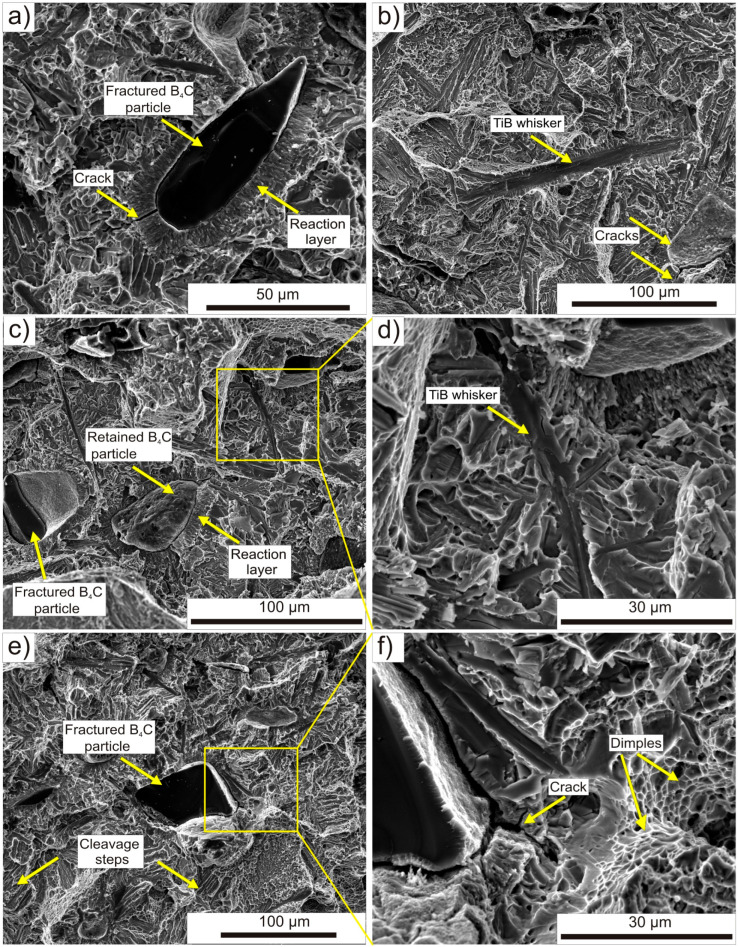
SEM images of characteristics fracture surfaces after tensile tests of TMC: (**a**) surface with fractured B_4_C particle, (**b**) surface with visible TiB whiskers, (**c**) surface with accumulation of B_4_C particles and (**d**) close-up of marked zone, (**e**) surface with fractured B_4_C particle with fractured reaction layer and (**f**) close-up of marked zone.

**Figure 12 materials-16-07438-f012:**
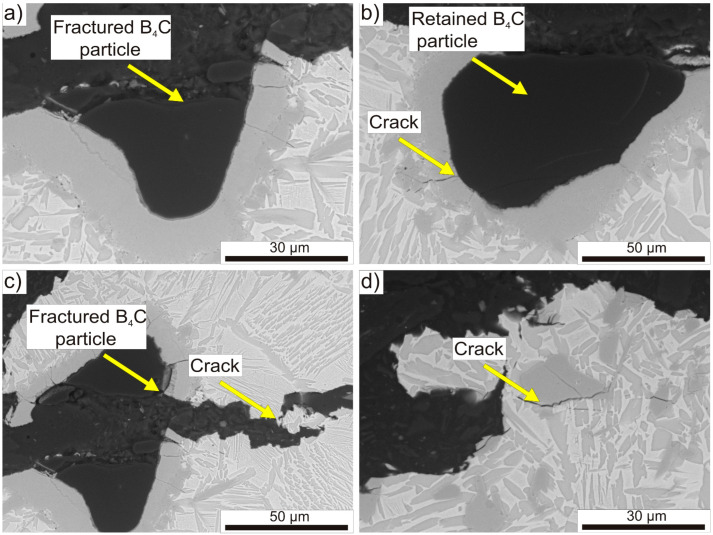
SEM images of the microstructure near the fracture surface of the lengthwise section of the tensile specimen: (**a**) fractured B_4_C particle, (**b**) retained B_4_C particle with fractured reaction layer, (**c**) fractured B_4_C particle with crack propagation into the matrix, (**d**) fracture at the interfaces between TiC phase and matrix.

## Data Availability

The data presented in this study are available upon request from the corresponding author.

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
