# Peer review of "Microstructure and Mechanical Properties of In Situ Synthesized Metastable β Titanium Alloy Composite from Low-Cost Elemental Powders"

_materials, 2023, doi:10.3390/ma16237438_

Round 1

Reviewer 1 Report

Comments and Suggestions for Authors

Dear author(s), please find below suggestions that may justify my final evaluation of the reviewed manuscript ‘Microstructure and mechanical properties of in-situ synthesized metastable β titanium alloy composite from low-cost elemental powders, Manuscript ID: materials-2720841.

The novelty of the work is lacking as most of the results are published in https://doi.org/10.3390/ma15175800 except tensile testing. The improvement in results has already been reported by https://doi.org/10.1016/j.matdes.2004.11.030.

Additionally some of the additional comments for authors:

1.     The abstract could be improved by including comparative results with modified and original alloys.

2.     The introduction lacks a clear research aim.

3.     By outlining the present gaps and constraints in the field of titanium matrix composites research and describing its current condition, the introduction might give greater perspective.

4.     A thorough examination of pertinent previous research is necessary to establish the study's context and justification. There is only one paper from 2022 and none from 2023. Please add any of the numerous articles that are available on the subject of the study as well as the mechanical properties and processing of beta titanium alloys.

5.     The study lacks specific details on the reaction mechanism, the formation of reinforcing phases (TiBx and TiC), and their impact on the microstructure and mechanical properties.

6.     Further details about the kinetics and thermodynamics of the in-situ process, as well as the variables affecting the precipitation of strengthening phases, would be beneficial.

7.     It would be helpful to include more information about the consolidation process, such as the hot compaction temperature, pressure, and duration.

8.     the conclusion does not offer a critical evaluation of the research's shortcomings or prospective directions.

Reviewer 2 Report

Comments and Suggestions for Authors

The paper concerns production of a strengthened beta titanium alloy by addition of a small percentage of boron carbide followed by hot compaction. The microstructure of the strengthened alloy is studied and phases of TiB, TiB2, TiC and B4C are identified. The characterization is done by XRD, optical microscopy, SEM and EDS mapping. Electron back scatter images are obtained for composition mapping as well. The mechanical properties of the alloy strengthened with BC are enhanced over that of untreated alloy. Some edits are needed:

line 327 - this paragraph seems left over from an earlier draft.

Additional discussion of the color schemes of Figures 6 and 7 would be helpful, explain the colors and what is learned from their distribution in more detail

Discuss relative prevalence of phases around the B4C particles

Discuss if the width and structure of the reaction zone is expected to vary with treatment time

Reviewer 3 Report

Comments and Suggestions for Authors

This manuscript presents an experimental study regarding the fabrication of titanium alloy composite material. This work is generally well-written and clear, providing the necessary information regarding the properties of the produced material. However, it is required that some modifications are performed to the manuscript, before it can be considered for publication:

The authors should mention the standards used for testing of the properties of the produced composite material. The authors should also justify the values selected for test speed.

How was the average reaction layer determined?

Regarding the tests of mechanical properties, the authors should mention the number of repetitions of the tests and present all the relevant graphs in Figure 5a or an additional one, in order to indicate the variation of mechanical properties of the produced material.

In general, the discussion of the results is clear and well justified, based on various measurements and observations. However, there is a lack of in-depth investigation on the effect of reinforcement percentage or other parameters. It would be more interesting if the authors had performed an investigation on some additional parameter of the process as well, in order to determine the optimum process conditions for obtaining the best possible parameters for the composite material, instead of analyzing only a single experiment.

Reviewer 4 Report

Comments and Suggestions for Authors

The article entitled “Microstructure and mechanical properties of in-situ synthesized metastable β titanium alloy composite from 3 low-cost elemental powders” is an interesting one and sound scientific contributions. Though, the authors have to address on the following points before it is being considered for its publications.

1. The authors have developed Titanium based composite via hot compaction process and studied the mechanical properties of developed composites. However, why the authors didn’t compare with the based Ti-5Al-5Mo-5V-3Cr alloy in the same processing condition so that we can understand the role of TiC / B4C compared to unreinforced one?. This is the major concern in this article. If they have not carried out mean, they may compare with the others work and discuss the improved properties of their composite. It has to be addressed in the revised version.

2. In the abstract, the authors have mentioned the value of tensile strength and compressive strength. Here, the error value is to be represented so that we can understand the deviation

3. In the introduction, at the end of second and third paragraphs, some references are to be cited

4. The main objectives of the present research work are to be added at the end of introduction part

5. The schematic diagram of present research work including the raw materials, blending/mechanical alloying, hot-pressing, characterization, and mechanical behaviour is to be introduced in the materials & method section so that it will attract the readers.

6. Either in introduction part or materials & method section, the selection of hot pressing compared to spark plasma sintering is to be discussed as few lines

7. Based on Figure 3, the authors have highlighted the formation of TiC particles and TiC whiskers which is good. However, here, the authors have to carry- out EDS on the spot so that it can be confirmed.

8. The scale marker is missing in Figure 4.

Round 2

Reviewer 1 Report

Comments and Suggestions for Authors

Dear Authors, Thanks a lot for incorporating all suggestions and except point No 1, However, based on your request, I will accept it in its current form and wish you all the best for your future project.

Kind Regards,

Author Response

Dear Reviewer,

First of all, we would like to thank you again for all your valuable comments and suggestions.
We would also like to thank you for accepting our responses and for agreeing to the explanation regarding comment No. 1. 
We thank you for your kind words and also want to wish you all the best in your scientific research work.

Best regards,

on behalf of the co-authors, Corresponding author:

Krystian Zyguła PhD, MEng
Assistant Professor
AGH University of Cracow
Faculty of Metals Engineering and Industrial Computer Science
Department of Metal Forming and Metallurgical Engineering
Al. Mickiewicza 30
30-059 Kraków, PL
tel: +48 12 617 38 72
e-mail: [email protected]
www.plastmet.agh.edu.pl

Reviewer 3 Report

Comments and Suggestions for Authors

The authors performed some modifications to their manuscript. However, they should add the information about which standards e.g. ASTM or ISO were used for the tests, as well as about the procedure for measuring the reaction layer.

Author Response

Dear Reviewer,

We would like to express our gratitude for considering our responses to the previous review. We also appreciate the additional suggestions provided. We believe they will contribute to a clearer presentation of the results and enhance their understanding.

In the revised version of the manuscript, we have included additional information about the standard applied during the tests (lines 176-179 in the clean version) and details on the measurement method of the reaction layer (lines 234-239 in the clean version).

Best regards,

on behalf of the co-authors, Corresponding author:

Krystian Zyguła PhD, MEng
Assistant Professor
AGH University of Cracow
Faculty of Metals Engineering and Industrial Computer Science
Department of Metal Forming and Metallurgical Engineering
Al. Mickiewicza 30
30-059 Kraków, PL
tel: +48 12 617 38 72
e-mail: [email protected]
www.plastmet.agh.edu.pl